# Fatty Hepatocytes Induce Skeletal Muscle Atrophy In Vitro: A New 3D Platform to Study the Protective Effect of Albumin in Non-Alcoholic Fatty Liver

**DOI:** 10.3390/biomedicines10050958

**Published:** 2022-04-21

**Authors:** Francesco De Chiara, Ainhoa Ferret-Miñana, Juan M. Fernández-Costa, Alice Senni, Rajiv Jalan, Javier Ramón-Azcón

**Affiliations:** 1Biosensors for Bioengineering Group, Institute for Bioengineering of Catalonia (IBEC), The Barcelona Institute of Science and Technology (BIST), Baldiri I Reixac 10-12, 08028 Barcelona, Spain; aferret@ibecbarcelona.eu (A.F.-M.); jfernandez@ibecbarcelona.eu (J.M.F.-C.); alice.senni@outlook.it (A.S.); jramon@ibecbarcelona.eu (J.R.-A.); 2UCL Institute of Liver and Digestive Health, University College London, London NW3 2QG, UK; rjalan@ucl.ac.uk; 3ICREA-Institució Catalana de Recerca i Estudis Avançats, 08010 Barcelona, Spain

**Keywords:** crosstalk, tissue engineering, ammonia, 3R, NEFAs

## Abstract

The liver neutralizes endogenous and exogenous toxins and metabolites, being metabolically interconnected with many organs. Numerous clinical and experimental studies show a strong association between Non-alcoholic fatty liver disease (NAFLD) and loss of skeletal muscle mass known as sarcopenia. Liver transplantation solves the hepatic-related insufficiencies, but it is unable to revert sarcopenia. Knowing the mechanism(s) by which different organs communicate with each other is crucial to improve the drug development that still relies on the two-dimensional models. However, those models fail to mimic the pathological features of the disease. Here, both liver and skeletal muscle cells were encapsulated in gelatin methacryloyl and carboxymethylcellulose to recreate the disease’s phenotype in vitro. The 3D hepatocytes were challenged with non-esterified fatty acids (NEFAs) inducing features of Non-alcoholic fatty liver (NAFL) such as lipid accumulation, metabolic activity impairment and apoptosis. The 3D skeletal muscle tissues incubated with supernatant from fatty hepatocytes displayed loss of maturation and atrophy. This study demonstrates the connection between the liver and the skeletal muscle in NAFL, narrowing down the players for potential treatments. The tool herein presented was employed as a customizable 3D in vitro platform to assess the protective effect of albumin on both hepatocytes and myotubes.

## 1. Introduction

Two-dimensional cell experiments are still the gold standard for the in vitro applications in drug discovery and development. Along with animal models, they have been valuable for massive drug screening programs, narrowing down potential candidates for many diseases. However, the success rate of clinical trials is only 12% over 20 years with an average cost of $172 million/trial [1]. Providing better models that recapitulate diverse clinical phenotype is highly needed.

In the era of precision medicine, tissue engineering represents an incredible asset to dissect the mechanism(s) underlying complex diseases. Moreover, it is a promising tool to shed light on the organ-to-organ crosstalk, allowing a more efficient drug design and avoiding the use of animals for research.

In the last decades, Non-alcoholic fatty liver disease (NAFLD) has become the world’s leading cause of liver disease, paralleling obesity, and diabetes, and the third most common cause for need of liver transplantation [2]. NAFLD is a systemic disease characterized by an intracellular high fat content within the liver in absence or moderate alcohol intake. Histologically, it ranges from simple steatosis, where >5% of the hepatocytes contain lipid droplets, to Non-alcoholic steatohepatitis (NASH) characterized by liver inflammation and fibrosis. The main risk factors associated with fatty liver are dyslipidemia, insulin resistance, type 2 diabetes mellitus (T2DM), hypertension, and obesity [3]. The World Health Organization (WHO) reported that 1.9 billion of the global adult population (39%) are overweight. Of these, over 650 million are obese (13%) [4]. In 2019, 38 million children under age 5 years and 340 million adolescents were overweight or obese [4]. Cirrhosis is the first cause of death in NAFLD patients (44%), followed by cardiovascular disease (9%) and T2DM [5].

The consumption of junk food meals and sedentary lifestyle are the main causes for the epidemic obesity in western countries [6,7]. The unhealthy diet, in combination with lack of physical activity led to the increased association between fatty liver and sarcopenia defined as loss of skeletal muscle tissue, strength, and contractile function [8,9,10]. Between 20% and 40% of NAFL patients suffer from sarcopenia, a percentage that increases to between 30% and 60% in NASH patients [11]. The concurrent presence of NAFLD and sarcopenia led to a far worse dysglycemia and insulin resistance [12]. Sarcopenia, in turn, is a strong predictor of poor post-liver transplant outcomes ranging from a longer hospital stay, need for mechanical ventilation, intensive care, an increased risk of infection, etc., [13,14]. The potential targetable actors or at what stage of NAFLD the loss of skeletal muscle starts the vicious circle are still matters of debate.

Infusion of human serum albumin (HSA) reduces renal dysfunction, hospital readmissions and mortality in patients with decompensated cirrhosis [15,16]. HSA is synthesized exclusively by hepatocytes and released into the bloodstream, where it acts as a transporter of fatty acids. In a previous work, we have demonstrated that hepatocytes cultured in vitro release albumin in response to a lipid challenge for 48 h [17]. The hypothesis is that the increased albumin might be the biological compensation of the hepatocytes to the massive presence of extracellular lipids.

In this study, we cultured hepatocytes with skeletal muscle cells under a high-fat regimen using the traditional 2D culture system to reproduce some of the features of NAFL and sarcopenia in vitro. After that, we took advantage of the latest technology in tissue engineering to recreate in the laboratory the liver and the skeletal muscle in 3D, both in healthy and disease states, to confirm and amplify the effects of fatty hepatocytes on skeletal muscle cells (Figure 1A,B). In a previous work, we optimized a combination of gelatin methacryloyl (GelMA) and carboxymethyl cellulose methacrylate (CMCMA) that recreate an optimal 3D environment for long-term cell culture [18].The aims of this study are: (i) to evaluate the effect of fatty liver on skeletal muscle tissue; (ii) to reproduce the common features both of NAFL and sarcopenia using miniaturized tissues in 3D to reduce the use of animals; (iii) finally, to test the beneficial effect of albumin on fatty hepatocytes.

## 2. Materials and Methods

### 2.1. Cell Culture

AML12 (ATCC^®^ CRL-2254™, Manassas, VA, USA) is a mouse hepatocyte cell line established from the CD1 strain (line MT42) and transgenic for human TGF alpha. The cells were cultured in Dulbecco’s modified Eagle media (DMEM) at 37 °C in 5% CO_2_. DMEM:F12 media (ATCC 30-2006) was supplemented with 10% fetal bovine serum (FBS; ATCC 30-2020), 10 µg/mL insulin, 5.5 mg/mL transferrin, 5 ng/mL selenium, 40 ng/mL dexamethasone and 1% Penicillin-Streptomycin (growth media). C2C12 (ATCC^®^ CRL-1772™) is a mouse myoblast cell line established from myogenic cells isolated from dystrophic mouse muscle. DMEM (ATCC^®^ 30-2002™) supplemented with 10% FBS and 1% Penicillin-Streptomycin was used for C2C12 cells culture as growth media. For cell subculture, trypsin-EDTA (25200072, ThermoFisher, Washington, DC, USA) and phosphate-buffered saline (PBS) were used. For AML12 cells, cell media was changed to DMEM with 2% FBS (differentiation media) 24 h before the experiments and switched to phenol red-free DMEM with 2% FBS and 1% Pen Strep for fluorescence assays. For C2C12 cells differentiation, cell media was changed to DMEM supplemented with 2% horse serum (16050130, ThermoFisher, Washington, DC, USA) and 1% Penicillin-Streptomycin (differentiation media) to induce the differentiation of myoblasts into myotubes. For the experiments performed in 2D, the cell density was 2.6 × 10^4^ cells/cm^2^ for both cell lines.

### 2.2. Cell Encapsulation

The hydrogel employed for liver and skeletal muscle tissues was the same in composition and is published elsewhere [18]. Briefly, the hydrogel was a mixture of gelatin methacryloyl (GelMA, 5% w/V) and carboxymethyl cellulose methacrylate (CMCMA, 1% w/V), a biodegradable and non-biodegradable material, respectively, (e.g., for preparing 500 µL of prepolymer solution, 0.05 g of GelMA and 0.01 g of CMCMA were weighted). The mixture was incubated in 400 µL of culture media at 65 °C for 3 h and then sterilized with UV for 15 min. After complete pre-polymer dissolution, lithium phenyl(2,4,6-trimethylbenzoyl)phosphonate (LAP, 0.1% w/V) was added to the GelMA-CMCMA mixture as photo-initiator in culture media. Previous experiments were performed to establish the optimal cells count for each tissue as follows: AML12 cell density: 1.5 × 10^7^/mL; C2C12 cell density: 2.5 × 10^7^/mL, data not shown. Then, cells were combined with the GelMA-CMCMA mixture (1:1) and exposed to UV for 30 s. From now on, the pre-polymer and cells mix will be referred to as liver and skeletal muscle hydrogels. To give a specific shape to the hydrogels, polydimethylsiloxane (PDMS) stamps were used (Appendix A). The shape of the stamps is crucial to promote the differentiation of both hepatocytes and myoblasts. Hepatocytes grow better forming clusters; therefore, the stamps used for culturing hepatocytes were ring-shaped (8 mm in diameter). Differently, the skeletal muscle cells differentiate and fuse into myotubes when seeded into channel-shaped hydrogels that allow the correct alignment and growth (Appendix A) [19]. The plate used for the hydrogel fabrication was a standard 24-wells plate. After encapsulation, the liver and the skeletal muscle hydrogels were cultured with growth media for 1 day and then cultured with differentiation media for 3 days. Fresh media was replaced every day. Liver hydrogels were kept in culture for 30 days and skeletal muscle cells for 1 week.

### 2.3. Non-Esterified Fatty Acids Preparation

Non-esterified fatty acids (NEFAs) are an important metabolic fuel. Inappropriately elevated plasma NEFAs concentrations may have several adverse effects on both carbohydrate and lipid metabolism. To recreate an in vitro model that can recapitulate the feature of NAFL, palmitic acid and oleic acid were chosen. Palmitic acid is a saturated fatty acid, while oleic acid is a monounsaturated fatty acid. NEFAs were resuspended in isopropanol due to their insolubility in water solutions. To dissolve them in the medium and allow them to cross the cellular membrane, the NEFAs need a carrier. The most used carrier is albumin, which binds free fatty acids and is internalized by the cells. Bovine serum albumin (BSA) fatty acid free (A8806, Sigma, Irvine, Germany) solution was prepared adding 7 g of BSA to 100 mL of PBS (1 mM, stock solution) in a pre-warmed 37 °C water bath on a heated stir plate. Once dissolved, the solution was filtered and aliquots were stored at −20 °C. Palmitic acid (PA) (P0500, Sigma) solution was prepared adding 1 g of PA to 9.75 mL of 100% isopropanol (400 mM, stock solution) and aliquots were stored at −20 °C. Oleic acid (OA) (O1008, Sigma) solution was prepared adding 1.2 mL of OA to 8.25 mL of 100% isopropanol (400 mM, stock solution) and aliquots were stored at −20 °C. NEFAs/BSA molar ratio was kept at 4:1 (From now on, 400 μM:100 μM will be the 1× stock solution), which represents a suitable equilibrium to mimic the pathological situation.

### 2.4. CellTiter 96^®^ Aqueous One Solution Cell Proliferation Assay (MTS)

To assess the cell viability, MTS (G3582, Walldorf, Germany) was added to the wells where the cells were previously seeded keeping the media: MTS ratio at 1:5. For example, in a 96-well plate, 20 µL of MTS were added to 100 µL/well of fresh media and incubated at 37 °C for 2 h in a humidified, 5% CO_2_ atmosphere. Absorbance at 490 nm was used to measure the amount of soluble formazan produced by cellular reduction of MTS.

### 2.5. AdipoRed™ Assay

To measure the accumulation of intracellular triglycerides, AdipoRed™ (PT-7009, Lonza, WL, Germany) solution was employed. The volume used varied depending on the plate size. For example, in 96-well plates, the volume of AdipoRed™/well was 5 µL in 195 µL of phenol red-free media. After each experiment, the plates were removed from the incubator and assayed for intracellular triglyceride content. The cells were rinsed twice with 200 µL of PBS to remove any residues of media. The plates were incubated for 10 and 30 min according to 2D and 3D experimental setups, respectively. Finally, the fluorescence was measured in the fluorimeter with excitation at 485 nm and emission at 572 nm and pictures were taken using an inverted microscope.

### 2.6. Neutral Red Assay

To assess the lysosomal arrangement of the cells, 4 mg/mL of Neutral red (N4638, Sigma-Aldrich, Milan Italy) were dissolved in 10 mL of PBS and filtrated (stock 1000X). One day before the assay, the stock was diluted in serum-free media (1X) and left in the incubator overnight. Doing this, the neutral red saturates the media, and any excess of the dye is removed the following day by centrifugation (10 min at 1800 rpm). The experiments were performed in 2D setup. On the day of the assay, the cells were washed twice with PBS and incubated with media containing neutral red for 3 h at 37 °C, 5% CO_2_. After that, the cells were washed again with PBS and images were taken using an inverted microscope. Finally, the neutral red was extracted from the cell using a de-staining solution made of 50% ethanol 96%, 49% H_2_O and 1% glacial acetic acid. The absorbance was recorded at 540 nm.

### 2.7. Oil Red O Staining

On the day of the experiment, the cells were washed twice with PBS, followed by fixation with 1 mL of formalin for 15 min under the fume hood. After that, two washes with deionized water were made. Oil Red O (O1391, Sigma-Aldrich, Milan, Italy) was diluted with ethanol 96% at a 3:2 ratio with water. Then, 1 mL of Oil Red O was added to each well. Samples were washed with Milli-Q water until the solution became clear. After that, 1 mL of Harris Hematoxylin solution (HHS16-500ML, Sigma-Aldrich, Milan, Italy) was added for 1 min and then washed extensively with Milli-Q water. Finally, 1 mL of Milli-Q water was added to each well and pictures were taken using an inverted microscope.

### 2.8. Ammonia, Albumin, ALT, and AST Measurements

The supernatants were frozen at −80 degrees until the measurements were performed. Roche Cobas c 311/501 analyzer was employed for all the measurements. Ammonia, Albumin, Alanine Aminotransferase (ALT), and Aspartate Aminotransferase (AST) were quantified using NH3L2, ALB2, ALTL, and ASTL cartridges based on enzymatic assays, respectively.

### 2.9. Live/Dead Staining

For each test, ethidium homodimer-1 (2 μM) (L3224, TermoFisher, Washington, DC, USA) and calcein-AM (4 μM) were added to PBS and mixed according to the manufacturer’s protocol. Afterward, the plate was incubated for 30 min and rinsed with PBS three times. The hydrogels were observed under confocal microscopy.

### 2.10. Hematoxylin and Eosin Staining

The experiments were performed in 2D. At end of the experiment, the samples were washed three times with PBS followed by formalin fixation for 10 min under the fume hood. After that, three more washes with PBS were undertaken and the samples were kept a 4 °C until further processing. The PBS was removed by Milli-Q water washes and the samples were incubated with Hematoxylin (H9627-100G, Sigma-Aldrich, Milan, Italy) for 3 min. To allow the stain to develop, the sample passed through deionized and tap water. Subsequently, the staining excess was washed out with a quick dip in acid ethanol (1 mL concentrated HCl + 400 mL of ethanol 70% in water). The samples were washed twice with tap and deionized water, and were incubated with Eosin Y solution (HT110116-500ML, Sigma-Aldrich) for 30 to 45 s. Then, the samples were washed with Milli-Q water. Pictures were taken using an inverted microscope.

### 2.11. Scanning Electron Microscopy

Encapsulated hepatocytes and myoblasts were washed twice with 0.1 M PBS pH = 7.4 for 10 min and then fixed in a 2.5% glutaraldehyde solution in 0.1 M PBS pH = 7.4 for 1 h. After that, samples were washed five times with PBS and kept at 4 °C. Samples were subsequently dehydrated through stepwise incubation in a series of graded ethanol baths at 50% (once for 10 min), 70% (twice for 10 min), 90%, 96%, and 100% three times for 10 min each. Samples were submitted to critical point drying, and finally, they were coated with carbon and analyzed under the microscope.

### 2.12. Transmission Electron Microscopy

Encapsulated hepatocytes and myoblasts were washed twice with 0.1 M PBS pH = 7.4 for 10 min and fixed in a 2% paraformaldehyde + 2.5% glutaraldehyde solution in 0.1 M PBS pH = 7.4 for 30 min. After that, samples were washed with fixation solution and kept at 4 °C. Samples were washed three or four times for 10 min at 4 °C and then were put in 1% osmium tetroxide and 0.8% potassium ferricyanide for 2 h at 4 °C using the buffer solution employed for fixation. After this, samples were washed four times for 10 min at 4 °C and then were subsequently dehydrated with acetone at 4 °C (50% for 10 min, 70% overnight, 80% for 10 min, and 90%, 96%, and 100% for three 10 min repetitions). The inclusion of the samples in epoxy resin (Spurr) was performed at room temperature (RT) with agitation: three volumes of acetone per 1 volume of Spurr (overnight), two volumes of acetone per 2 of Spurr (6 h), 1 volume of acetone per 3 of Spurr (overnight) and pure Spurr (3 changes of 6 h, overnight and 4 h). The blocks were made and left for 72 h in the oven at 60–80 °C and then cut in semi-thin sections (survey sections) using a glass blade and an ultramicrotome and methylene blue as a contrast agent. Before performing the ultrathin sections, the area of interest was selected using an optical microscope. Ultramicrotomy was performed with a diamond blade, and 70 nm slices were collected on copper grids. Ultrathin sections were counterstained with uranyl acetate and lead citrate. The observation was made on a TEM JEOL J1010 using a Gatan Orius CCD camera with Gatan Microscopy Suite (GMS) software suite (DM3, Gatan, Pleasanton, CA, USA).

### 2.13. Immunofluorescence Staining

Samples were fixed in a 10% formalin solution, washed with PBS, and permeabilized with Triton X-100 0.1% in PBS for 10 min. Then, samples were incubated with different antibodies such as albumin antibody, OTC antibody, glutamine synthetase antibody, and MYH7 antibody in PBS overnight at 4 °C. The day after, the excess of the antibody was washed away with PBS, and the samples were then incubated with rhodamine-phalloidin and with different antibodies such as goat anti-rabbit, goat anti-mouse in blocking solution for 2 h. Nuclei were stained with one µM DAPI for 30 min (Appendix A).

### 2.14. Real-Time qPCR

Two-steps protocol was employed to isolate total RNA from cells inside the hydrogels. First, the hydrogels were soaked in 1 mL TRIzol™ (15596026, Thermofisher, Washington, DC, USA and smashed using TissueLyser LT (Qiagen, Hilden, Germany). After that, 200 μL of chloroform was added to the mixture and incubated for 5 min at room temperature. The samples were centrifuged at 10,000 rpm for 15 min. The upper phase was mixed with 500 μL of isopropanol and incubated for 10 min. One more centrifugation was made at 10,000 rpm for 10 min at 4 °C, and supernatant was removed. The pellet was then cleaned up using RNeasy kit (74004, Qiagen, Hilden, Germany). The RNA concentration and integrity of each RNA sample was determined with the NanoDrop 2000 and electrophoretic run-on agarose gel at 2% w/V. One µg of total RNA was retrotranscribed to cDNA using QuantiTect Reverse Transcription Kit (205311, Qiagen, Hilden, Germany). One µL of cDNA was used to set up the amplification reaction using either SYBR green or TaqMan chemistries. Each sample was tested in quadruplicates. Target genes for AML12 are *Gapdh*, *Hgf*, *Hnf4*a, whereas C2C12 are *Gapdh*, *MyoG*, *Tnnt1*, *Foxo3*, *Fbx32* (Appendix A). Target genes were normalized using beta-actin as endogenous control and their relative quantification was carried out with 2^−ΔΔCt^ method (where Ct represents the threshold cycle) using the untreated cells as calibrator. The amplification efficiency of target and reference genes was approximately the same (slope < 0.1).

### 2.15. Pre-Treatment of Hepatocytes with Albumin

AML12 cells were pre-treated with 4% w/V Human Serum Albumin (Sigma A3782–100MG, Milan, Italy) for 1 h prior to exposure to NEFAs [15]. Three washes with PBS were performed to remove the excess of albumin from the media.

### 2.16. Statistics

Results are expressed as mean values ± SEM and compared using one-way analysis of variance followed by Dunnet’s or Tukey’s multiple comparison post hoc tests, where appropriate. *p* values ≤ 0.05 were considered significant. Results were analyzed using GraphPad Prism software (GraphPad 8.0, La Jolla, CA, USA).

## 3. Results

### 3.1. Hepatocytes Exhibit Lipid Accumulation upon Treatment with Non-Esterified Fatty Acids in 2D Culture

To induce lipid accumulation within the hepatocytes, we first tested palmitic/oleic acids combinations (PA/OA) at different ratios for 48 h. We found that the mixture of PA/OA (from here onwards, it will be defined as NEFAs solution) at ratio 2:1 induces the highest lipid internalization as demonstrated by Oil-red-O staining (green line, Figure 2A, Appendix A). Furthermore, only the treatment of NEFAs at 400 µM reduces the metabolic activity at all time points under investigation (*p* < 0.0001, *p* < 0.001, and *p* < 0.05, respectively, Figure 2B). To confirm the intracellular lipid accumulation upon treatment with NEFAs, we employed the AdipoRed™ assay. We observed an increase in intracellular lipids at 72 h after treatment with 100 and 400 µM condition compared with the untreated condition, although it is not significant (*p* = 0.0577) (Figure 2C). Next, we used the Neutral red assay to check the lysosomal arrangement upon treatment with NEFAs [20]. We observed an increase in the absorbance due to NEFAs treatment (400 µM) compared to the untreated condition at 24, 48 and 72 h (*p* < 0.01, *p* < 0.001, and 0.0541, respectively, Figure 2D). At the microscopic examination, fewer cells were present in NEFAs treated condition (400 µM), but they were full of acidic bodies surrounding the nuclei compared to the untreated cells (Figure 2D). In the supernatant of hepatocytes treated with 400 µM of NEFAs solution, we found higher levels of ammonia, ALT, AST, and free-DNA. In contrast, lower levels of albumin compared to the untreated condition after 72 h of treatment were observed (Figure 2E). We did not find any difference in the urea production among all the conditions under investigation (Appendix A). Taken together, these data suggest an impairment of hepatocytes’ functionality caused by NEFAs treatment with consequent release in the media of ammonia, ALT, AST, and free-DNA.

### 3.2. The Media from Hepatocytes Induces Loss of Cytoplasm in Skeletal Muscle Cells in 2D Culture

The next step was to expose skeletal muscle cells to the media from AML12 treated or untreated with NEFAs to see the effect on the cell phenotype. Briefly, after 4 days with differentiation media from the encapsulation, C2C12 cells were treated with the following conditions: fresh differentiation media (untreated), supernatant collected from healthy AML12 cells-untreated hepatocytes’ supernatant (HS) -, and the supernatant collected from AML12 cells treated with 100 µM and 400 µM NEFAs (treated hepatocytes’ supernatant (THS) 100 and 400 µM, respectively). The MTS assay showed reduced metabolic activity in Treated HS 400 µM condition at 24, 48, and 72 h (*p* < 0.05, *p* < 0.001 and *p* < 0.001, Figure 2F) compared with untreated HS. At 72 h, Haematoxylin and eosin staining showed a reduction in the myotube size upon treatment with media coming from hepatocytes treated with the highest concentration of NEFAs. The reduction of eosin staining showed a loss of cytoplasm volume in treated myotubes (black arrows, Figure 2G). Interestingly, in the THS condition, fewer myotubes were present, but number of nuclei was not affected, suggesting that the supernatant from fatty hepatocytes affects the differentiation rather than viability. The loss of cytoplasm also occurred in the untreated C2C12 starting from 120 h of culture (Appendix A). Thus, it is not possible to further study this phenomenon using conventional 2D culture where many of the in vivo features are lost. We combined two 3D tissues to investigate the relationship between liver and skeletal muscle tissues upon NEFAs treatment.

### 3.3. Encapsulated Hepatocytes in 3D Hydrogels Are Viable and Express Specific Markers

To culture the hepatocytes in 3D, we used the GelMA-CMCMA composite hydrogel (see M and M section), previously published by our laboratory [19]. We used different sizes of the polymer to obtain the optimal nutrient diffusion, we exposed 150 and 50 µL of polymer containing 1.5 × 107 cells/mL to UV light and left in culture for up to 8 days (Figure 3A). At 1 and 8 days, we performed a live/dead assay and MTS to check the metabolic activity and temporal/spatial cell viability. The MTS assay showed higher metabolic activity in the 150 µL hydrogel compared with the 50 µL hydrogel (the cell density/mL was the same in both conditions) in both timepoints under investigation (Figure 3B). For further experiments, we have chosen 50 µL as the volume of encapsulation because of the higher live/dead cells ratio compared to 150 µL (Figure 3C,D and Appendix A, respectively). These data indicate that the number of live cells increases over time compared to dead ones in the smaller volume of the polymer. We next sought to evaluate whether the encapsulating process influences the expression of specific hepatocyte markers, such as ornithine transcarbamylase (OTC), glutamine synthetase (GS) (enzymes involved in the nitrogen metabolism), and albumin. The immunocytochemistry stainings showed the expression of those markers over 30 days (Figure 3E,F). The hydrogel allows the long-term culture of the cells without affecting their phenotype, unfeasible using 2D cultures.

### 3.4. 3D Hepatocytes Exhibit Pathogenic Lipid Accumulation upon Non-Esterified Fatty Acids Challenge

To evaluate, qualitatively and quantitatively, the intracellular fat in the 3D tissues, we used the NEFAs solution previously established in the 2D model and performed the AdipoRed™ assay (Figure 4A, Appendix A). We detected a considerable increase in intracellular lipids and cell size at 72 h after treatment in the 400 µM condition compared with the untreated condition. At the ultrastructural level, the protocol of scanning electron microscopy did not affect the hydrogel structure (Figure 4B top panel). At 650×, the cell membrane of untreated cells is regular. The cells tend to form clusters, whereas signs of membrane damage and cellular debris are visible upon treatment with NEFAs in a dose-dependent manner. At 2200×, in the NEFAs condition, the lipid accumulation caused an increase in cell size, cytoplasmic membrane disruption, and appearance of apoptotic bodies (Figure 4B and Appendix A). Transmission electron microscopy confirmed the accumulation of lipids and destruction of the plasmatic membrane (Appendix A). These data were confirmed by MTS assay, where the metabolic activity was affected in NEFAs-treated cells in a dose-dependent fashion (Figure 4C). To assess whether NEFAs treatment alters hepatocyte functionality, we analysed hepatocyte markers by gene expression such as hepatocyte growth factor (Hgf) and hepatocyte nuclear factor 4 alpha (Hnf4a). Since the glyceraldehyde 3-phosphate dehydrogenase (Gapdh) was affected by NEFAs treatment, we used beta-actin as housekeeping gene. The expression of Hgf measured by real-time qPCR was significantly higher in 100 and 400 µM NEFAs (*p* < 0.001 and *p* < 0.01, respectively). In contrast, the expression of Hnf4a was significantly decreased in 100 µM and 400 µM NEFAs (*p* < 0.001 and *p* < 0.0001, respectively) compared to the untreated cells. The gene Gapdh was significantly decreased in 100 µM and 400 µM NEFAs (*p* < 0.001 and *p* < 0.0001, respectively) compared to the non-treated condition. To see whether the data obtained in the 2D cultures correlated with the 3D model, we analysed the content of released factors from hepatocytes within the media (Figure 4D). We found a high level of ammonia (*p* < 0.001, Figure 4E) but an unchanged level of urea compared to untreated cells (Appendix A). Furthermore, a significant increase in ALT, AST (*p* < 0.01, both), and decreased level of albumin (*p* < 0.05) were found compared with the untreated condition (Figure 4E). We also measured the free-DNA content as a marker of cell death. In the highest NEFAs condition, we found that the free-DNA content was significantly higher than the untreated condition at 72 h (*p* < 0.001, Figure 4E). The 3D fatty hepatocytes generated significantly higher levels of ammonia, AST and free-DNA (*p* < 0.0001, *p* < 0.05, and *p* < 0.0001, respectively) than the 2D models, whereas there was no difference between untreatred groups. The data suggest that (i) NEFAs treatment impairs hepatocyte functionality; (ii) and that the cells cultured in 3D models have amplified response upon NEFAs treatment compared to 2D cultures.

### 3.5. 3D Fatty Hepatocytes’ Supernatant Affects Metabolic Activity and Differentiation State of 3D Skeletal Muscle Cells

We encapsulated the C2C12 myoblasts using the protocol previously developed in our laboratory to generate 3D muscle microtissues [19]. After 4 days of differentiation, the 3D muscles were treated with the following conditions up to 72 h: fresh differentiation media, supernatant collected from 3D healthy AML12 cells (untreated HS), cells treated with 100 µM and 400 µM NEFAs (THS 100 and 400 µM, respectively). Using confocal microscopy, we observed a reduction in both cytoplasm and differentiation markers such as myosin and F-actin in myotubes incubated with supernatant from fatty hepatocytes (THS 400 µM) compared to untreated condition (Figure 5A). As for 3D liver tissue, the hydrogels withstood the SEM process without losing their channel shape (Figure 5B, top panel). Neither the differentiation media nor untreated HS affected the differentiation of myoblasts into myotubes visible throughout the 3D structure (Figure 5B, middle and bottom panel). Remarkably, the myoblasts treated with THS showed a marked loss of cytoplasm and a decrease in myofiber diameter (Figure 5B). Furthermore, they present a rough surface compared to the smoother ones in the untreated and Untreated HS conditions. We next sought to determine the effect of the supernatant on the proliferation and viability of 3D myotubes and whether it follows a dose-dependent fashion. To this aim, we performed the MTS assay at 72 h. We observed a significant decrease in the absorbance in 400 µM NEFAs supernatant compared to the untreated (*p* < 0.01, Figure 5C). Remarkably, we observed no significant changes in the absorbance in the NEFAs condition compared to the untreated condition. Next, we acted to determine whether fatty HS treatment affects muscle maturation. We analysed gene expression of markers of different stages of myoblast differentiation using qRT-PCR. Additionally, in this case, the Gapdh was affected by NEFAs treatment; therefore, we used beta-actin as housekeeping gene. The Myogenin (MyoG) gene increased, while Troponin T1 (Tnnt1) gene expression decreased significantly upon NEFAs treatment (Figure 5D). To see whether the loss of cytoplasm and myofiber reduction is due to dedifferentiation or atrophy, F-Box Protein 32 (Fbx32) and Forkhead Box O3 (Foxo3) atrogenes were measured. Only the media containing the supernatant from the highest NEFAs content induced upregulation in Fbx32 and Foxo3 genes (Figure 5D). These results demonstrate the detrimental effects of the HS rich in ammonia, ALT, AST and free-DNA on the phenotype and functionality of myotubes.

### 3.6. Albumin Pre-Treatment Prevents Lipid Accumulation within the Hepatocytes Rescuing Skeletal Muscle Cells from Hepatocyte-Induced Damage in 3D Culture

In a previous study (17), we observed the increased albumin release from the hepatocytes upon challenge with NEFAs at 48 h while it dramatically decreases at 72 h of treatment (Figure 4E). This temporary response of the cells suggests that albumin might neutralize, somehow, part of the extracellular lipids. To test this hypothesis, we pre-treated the AML12 cells with 4% albumin for 1 h prior to treatment with NEFAs. To control the impact of albumin alone; we used fresh cell culture media with 4% albumin on untreated cells as a control group. We observed a significative reduction in ammonia, ALT, AST, free-DNA and an increase in albumin released in the albumin pre-treated condition compared to only NEFAs treatment (*p* < 0.01, *p* < 0.05, *p* < 0.05, *p* < 0.0001 and *p* < 0.05, Figure 6A). These protective effect results were confirmed also at mRNA level, where the Hgf, Hnf4, and Gapdh genes expression in the albumin pre-treated condition tend to the level of control (Figure 6B). Furthermore, the mitochondrial enzyme involved in the metabolism of ammonia, Ornithine transcarbamylase (OTC), was increased in the albumin pre-treated condition compared with NEFAs condition (Figure 6C, Appendix A). Furthermore, the intracellular lipids accumulation was reduced by albumin pre-treatment as shown by AdipoRed™ assay (Figure 6D, Appendix A). At SEM analysis, the cells pre-treated with albumin and then treated with NEFAs presented a phenotype similar to the control, preventing the cell membrane damage (Figure 6E).

When the conditioned media were used to treat skeletal muscle cells, the myotubes in the albumin pre-treatment condition were thicker than the THS 400 μM condition (Figure 7A). At the ultrastructural level, the myotubes were thicker in the albumin pre-treated THS 400 μM condition than the THS 400 μM (Figure 7B). Albumin also improves metabolic activity of 3D skeletal muscle tissues and normalizes the level of the genes involved in atrophy (Fbx32 and Foxo3, Figure 7C,D). Taken together, these results show that the hepatocytes damage induced by NEFAs was attenuated by the pre-treatment with albumin preventing the loss of skeletal muscle mass.

## 4. Discussion

We used engineered tissues to demonstrate incontrovertibly the direct cellular crosstalk between fatty hepatocytes and skeletal muscle cells in a novel in vitro model to study sarcopenia in NAFL. First, we proved the advantages of using 3D models to study the mechanism(s) behind complex diseases such as NAFLD over the conventional 2D cultures. Second, fatty hepatocytes release multiple chemical factors (e.g., ammonia) that induce atrophy in fully differentiated myotubes. Third, the albumin curbs the NEFAs’ induced damage to the hepatocyte and, as a consequence, to the myotubes.

We took advantage of the standardized combination of biodegradable GelMA and non-biodegradable CMCMA polymers to fabricate a long-lasting 3D cell structure (18). The hydrogels are highly customizable in shape, size and cell density, mimicking the native properties of the organs. Within the GelMA-CMCMA, the liver cells change from flat and stretched to more spheroidal and native phenotype (Figure 2C and Figure 4A, Appendix A), showing differences between the apical and basal side, as demonstrated by SEM images (Figure 4B). The photo-mold patterning technique provides to the skeletal muscle cells the physical constriction essential for the high degree of myotubes differentiation, unachievable in the 2D conventional culture method (Figure 2G and Figure 5B). Moreover, protein expression levels in 3D cultured cells resemble the levels found from cells in vivo, contrary to 2D cultures [21]. Our 3D cultures produced an amplified response to the NEFAs compared with 2D cultures, as shown by the level of ammonia and albumin (Figure 4E). The greatest benefit given by the 3D culture is the possibility to prolong experiments up to 30 days without affecting viability and/or differentiation otherwise impossible to achieve in 2D, where the myotubes differentiation is lost but not the viability (Figure 3E,F, Appendix A). Moreover, 3D cultures allow us to see dramatic changes of cellular phenotype upon treatment with NEFAs resembling the hepatocytes isolated from in vivo [22]. This kind of intracellular modification cannot be seen in a 2D model. Remarkably, signs of cell death such as membrane damage and apoptotic bodies, as well as cellular fragmentation, were observed in the hepatocytes due to the NEFAs treatment. As Feldstein et al. claims [23], hepatocyte apoptosis is significantly augmented in NASH patients and correlates with disease severity. At the gene expression level, *Gapdh* gene expression was downregulated, probably due to the shift from glucose to lipid metabolism in fatty hepatocytes. We found a high level of *Hgf* in a similar way that it has been found increased in serum of NASH patients, although is not known if that it is a consequence of fat accumulation or it is a compensatory process. For example, in the presence of oxidative stress due to a high fat regimen, *Hgf* has an antioxidant response [24]. On the contrary, we found the level of *Hfn4* decreased upon NEFAs treatment. The *Hfn4* gene activates hepatic gluconeogenesis [25], confirming the downregulation of glycolysis, and triggers insulin genes both directly and indirectly [26,27], regulating genes involved in the progression of NAFLD [28]. On the same line, the supernatant from 3D fatty hepatocytes contains high levels of ammonia and cell death components such as ALT, AST, and free-DNA, all markers of damaged and dead cells (Figure 4E) [29]. These components might interact with skeletal muscle cells and might play an important role in the pathologic phenotype under a high-fat regimen [30].

The treatment of the supernatant collected from fatty hepatocytes impaired the mass and function of 3D myoblasts. Data showed loss of cytoplasm, reduction of the metabolic activity and the myotube diameter, all features associated with loss of differentiation. At genes level, the expression of markers of myotube differentiation and atrophy were dysregulated. Particularly, the supernatant collected from fatty hepatocytes seems to induce a process of dedifferentiation as demonstrated by the overexpression of *MyoG* and under-expression of *Tnnt1*, the first involved in the early steps, whereas the second in the late stages of myotubes differentiation [31]. We further investigated the mechanism behind the loss of cytoplasm assessing the level of genes involved in atrophy. Specifically, the high levels of *Foxo3* and *Fbx32* demonstrated that the fatty hepatocytes trigger myotube reduction in size and function [32]. Atrophy has been found associated with high levels of myostatin, a potent negative regulator of skeletal muscle growth [33]. Qiu et al. recently showed that exposure of mouse skeletal muscle myotubes in culture to ammonium acetate caused a time- and concentration-dependent increase in myostatin mRNA and protein expression [34]. These data suggest that ammonia might be one of the main culprits for sarcopenia in NAFL.

This study presents some limitations. First, to optimize the shape, size, and stiffness of the biomaterial, we used mouse cell lines because of their cost and low variability between experiments. Future experiments will include human primary cells to better mimic the in vivo conditions. Second, we employed one liver cell type throughout the study. We did that to demonstrate the causality between fat accumulation within the hepatocytes and atrophy in skeletal muscle cells. However, we believe that hepatic stellate cells from the liver part and satellite cells from the skeletal muscle part play a crucial role in the interorgan crosstalk in NAFL, accelerating the progression of the disease faster and more aggressively. Third, the addition of a continuous flow has been demonstrated to better mimic the natural organ micro-environment. We decided not to include it to avoid additional variables that would have been difficult to control, adding variability between experiments. Fourth, we did not find any differences in urea in both 2D and 3D. Probably, some other compensatory mechanisms were in place (e.g., the sensitivity of the detection method was not sensitive enough to catch difference between the conditions under investigation).

Albumin’s biological effects span from binding of dangerous endogenous and exogenous ligands to plasma volume expander [35]. In this study, we confirmed the exceptional capacity of albumin to reduce or neutralize the destructive effects of lipids on the hepatocytes, mitigating the damage to the skeletal muscle engineered tissue. This might be possible in many ways, from facilitating the uptake of fatty acids by the cells to protecting mitochondria where lipid metabolism and urea cycle take place [36]. Future studies will be focused on two sides: (i) chemically modify albumin to enhance its beneficial effects; (ii) assess what other organs can be rescued by albumin pre-treatment followed by liver damage.

## 5. Conclusions

This study demonstrates the direct connection between the liver and skeletal muscle under a high fat regimen narrowing down the targeted players for potential future treatments. The tool herein presented was successfully employed as an in vitro platform for drug screening.

## Figures and Tables

**Figure 1 biomedicines-10-00958-f001:**
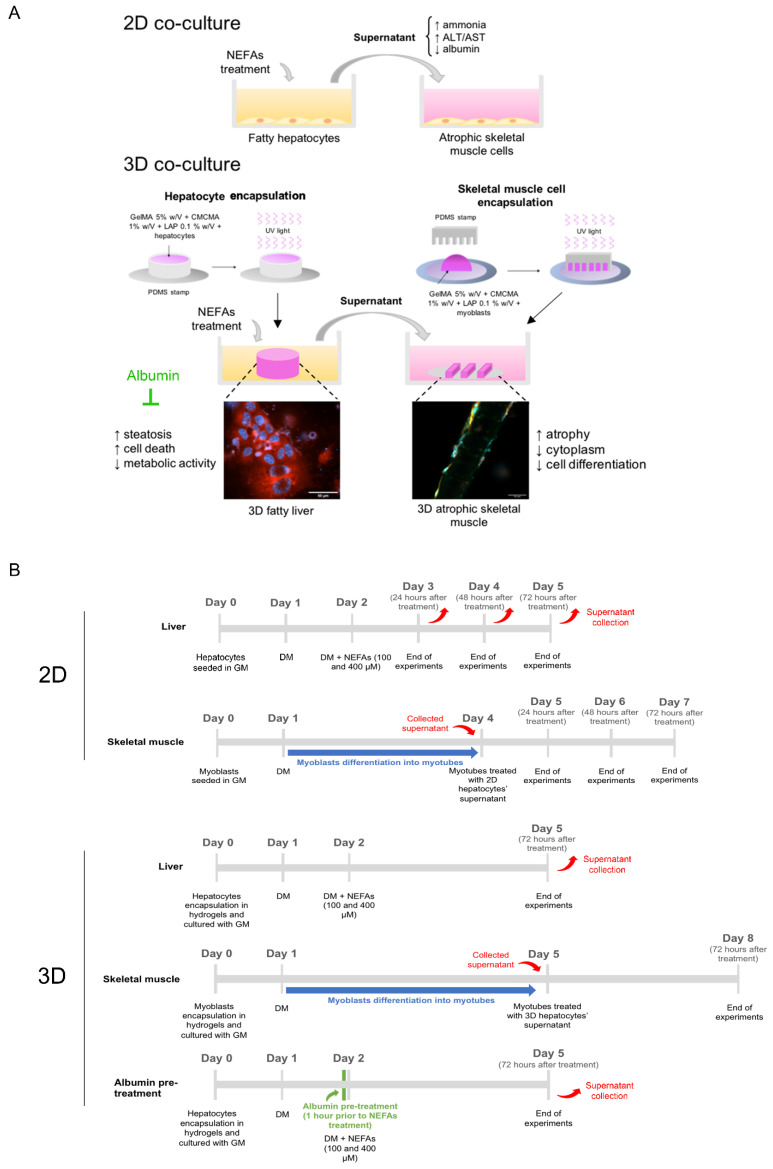
(**A**) Schematic representation of the in vitro experimental model employed in this study. (**B**) Schematic representation of the timeline of the experiments performed in this study. 2D: two-dimensional; NEFAs: non-esterified fatty acids; ALT: alanine aminotransferase; AST: aspartate aminotransferase; 3D: three-dimensional; GelMA: gelatin methacryloyl; CMCMA: carboxymethyl cellulose methacrylate; LAP: lithium phenyl(2,4,6-trimethylbenzoyl)phosphonate; UV: ultraviolet; PDMS: polydimethylsiloxane; GM: growth media; DM: differentiation media.

**Figure 2 biomedicines-10-00958-f002:**
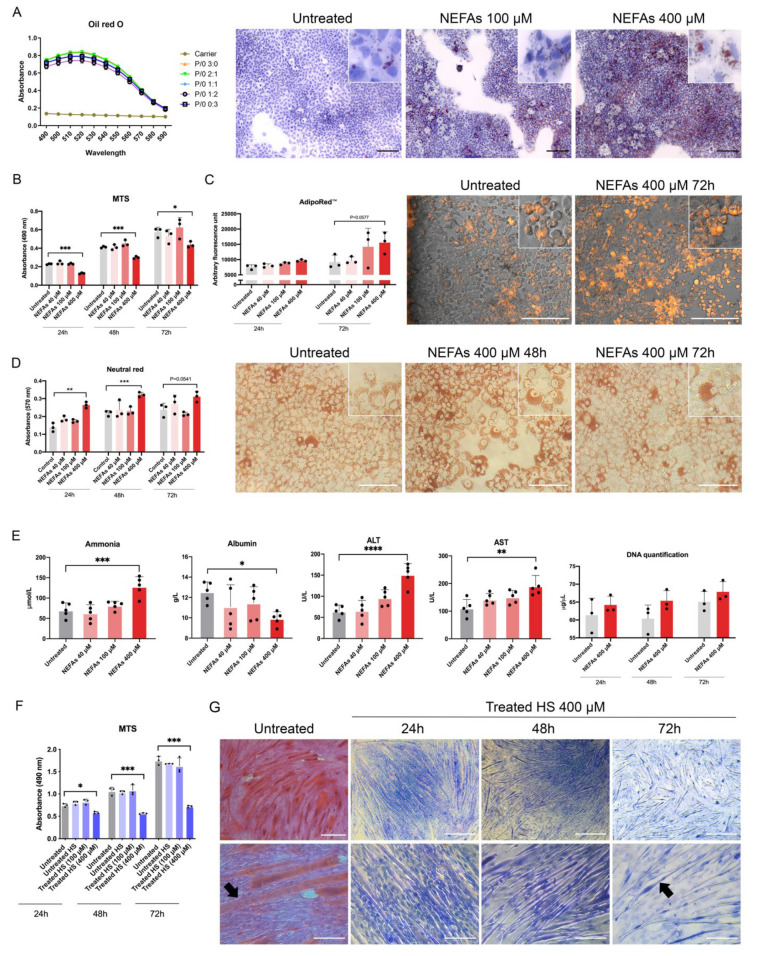
2D in vitro models of fatty hepatocytes (AML12) and sarcopenic skeletal muscle cells (C2C12). (**A**) Absorbance signal of various internalized non-esterified fatty acids (NEFAs) combinations assessed by Oil-red-O assay (P = palmitic acid; O = oleic acid) after 48 h. Scale bar = 500 µm. (**B**) Metabolic activity of AML12 cells challenged with various concentrations of NEFAs measured by MTS assay. (**C**) Intracellular lipid accumulation was assessed quantitatively (fluorimeter) and qualitatively (confocal microscopy) by AdipoRed™ assay. Scale bar = 100 µm. (**D**) AML12 cells’ lysosomal re-arrangement upon NEFAs challenge assessed quantitatively (spectrometer) and qualitatively (light microscopy) by Neutral red assay. Scale bar = 100 µm. (**E**) Biochemical analyses of the supernatant of AML12 challenged with NEFAs for 72 h assessed by clinical standard procedures. (**F**) Metabolic activity of C2C12 cells incubated with supernatant from fatty AML12 cells. (**G**) Haematoxylin and eosin staining of C2C12 cells incubated with supernatant from fatty hepatocytes. Black arrows indicate fused myoblasts (myotubes). Scale bar top panel = 400 µm, and bottom panel = 100 µm. The results are expressed as mean values ± SEM and compared using one-way analysis of variance followed by post hoc tests when appropriate. * *p* < 0.05; ** *p* < 0.01; *** *p* < 0.001; **** *p* < 0.0001. P: palmitic acid; O: oleic acid; NEFAs: non-esterified fatty acids; MTS: 3-[4,5,dimethylthiazol-2-yl]-5-[3-carboxymethoxy-phenyl]-2-[4-sulfophenyl]-2H-tetrazolium; ALT: Alanine Aminotransferase; AST: Aspartate Aminotransferase; HS: hepatocytes’ supernatant.

**Figure 3 biomedicines-10-00958-f003:**
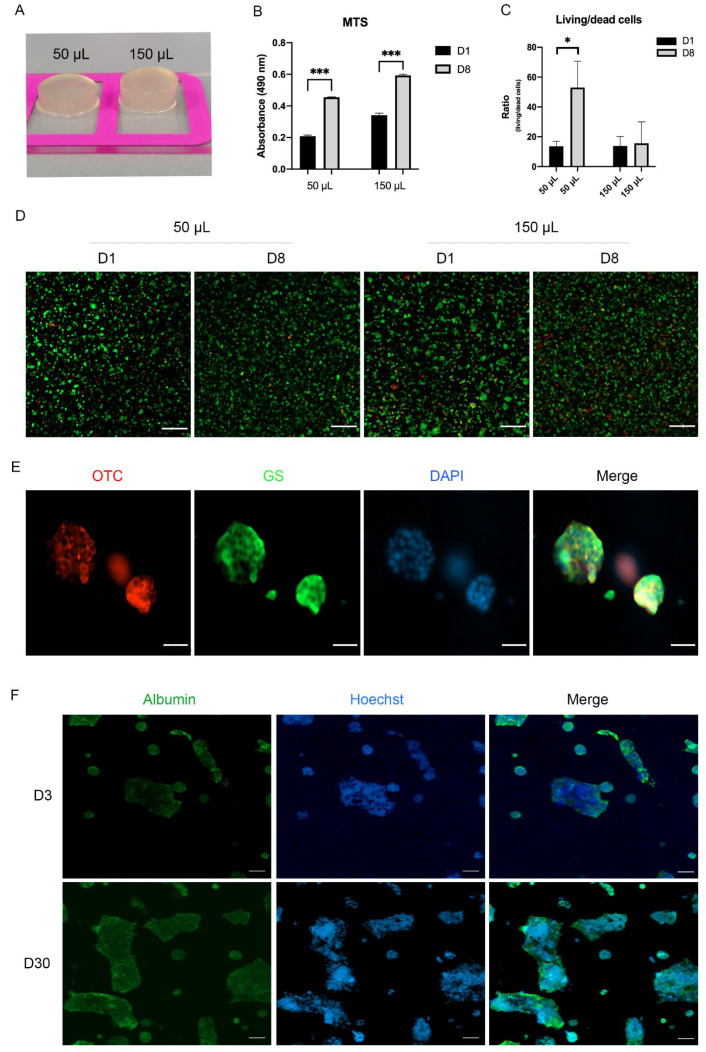
AML12 cell growth in 3D using GelMA-CMCMA polymer. (**A**) Picture of AML12 cell growth in 3D using GelMA-CMCMA polymer using two different volumes, cell density 1.5 × 107 cells/mL. (8 mm diameter) (**B**) Metabolic activity of encapsulated AML12 cells cultured over time. (**C**) Living-dead cells ratio obtained from 5 random z-Stacks/condition using live/dead assay. (**D**) Confocal images of live/dead assay of encapsulated AML12 cells. Living cells = green, dead cells = red. Scale bar = 200 µm (**E**) Confocal images of 3D AML12 cells expressing ornithine transcarbamylase (OTC) and glutamine synthetase (GS). Nuclei stained using DAPI. Scale bar = 50 µm. (**F**) Confocal images of 3D AML12 cells expressing albumin at indicated timepoints. Nuclei stained using DAPI. Scale bar = 50 µm. The results are expressed as mean values ± SEM and compared using one-way analysis of variance followed by post hoc tests when appropriate. * *p* < 0.05; *** *p* < 0.001. MTS: 3-[4,5,dimethylthiazol-2-yl]-5-[3-carboxymethoxy-phenyl]-2-[4-sulfophenyl]-2H-tetrazolium; OTC: ornithine transcarbamylase, GS: glutamine synthetase.

**Figure 4 biomedicines-10-00958-f004:**
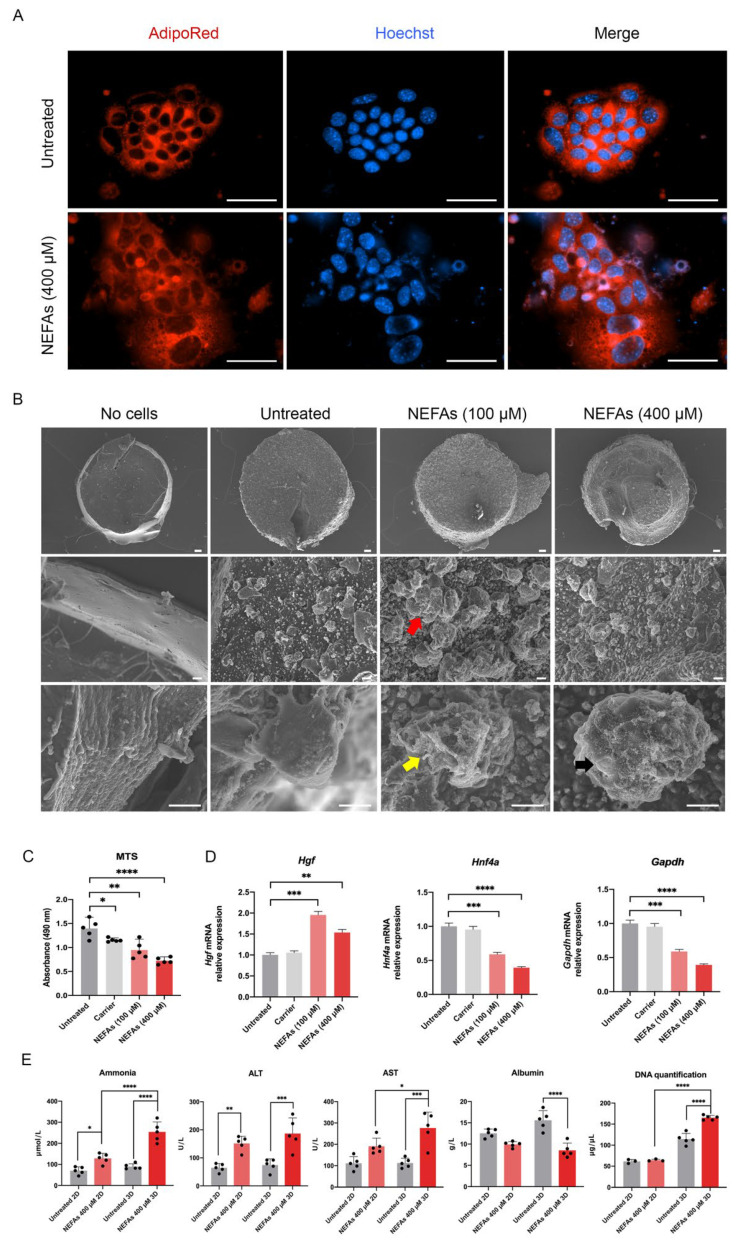
In vitro three-dimensional model of Non-alcoholic fatty liver. (**A**) Confocal images of lipid accumulation in NEFAs treated hepatocytes assessed by AdipoRed™ assay. Scale bar = 200 µm. (**B**) Ultrastructural assessment of AML12 cells phenotype by scanning electron microscopy. Red, yellow, and black arrows indicate increase in cell size, cytoplasmatic membrane disruption and apoptotic bodies, respectively. Scale bar top panel = 100 µm, middle panel = 10 µm, and bottom panel = 10 µm. (**C**) Metabolic activity of AML12 cells challenged with various concentrations of NEFAs measured by MTS assay. (**D**) Real-time qPCR of hepatocyte markers. The untreated condition was used as calibrator. (**E**) Biochemical analyses of the supernatant of 2D-3D AML12 challenged with NEFAs for 72 h assessed by clinical standard procedures. The results are expressed as mean values ± SEM and compared using one-way analysis of variance followed by post hoc tests when appropriate. * *p* < 0.05; ** *p* < 0.01; *** *p* < 0.001; **** *p* < 0.0001. NEFAs: non-esterified fatty acids; MTS: 3-[4,5,dimethylthiazol-2-yl]-5-[3-carboxymethoxy-phenyl]-2-[4-sulfophenyl]-2H-tetrazolium; Hgf: hepatocyte growth factor; Hnf4a: hepatocyte nuclear factor 4 alpha; Gapdh: glyceraldehyde 3-phosphate dehydrogenase; ALT: Alanine Aminotransferase; AST: Aspartate Aminotransferase.

**Figure 5 biomedicines-10-00958-f005:**
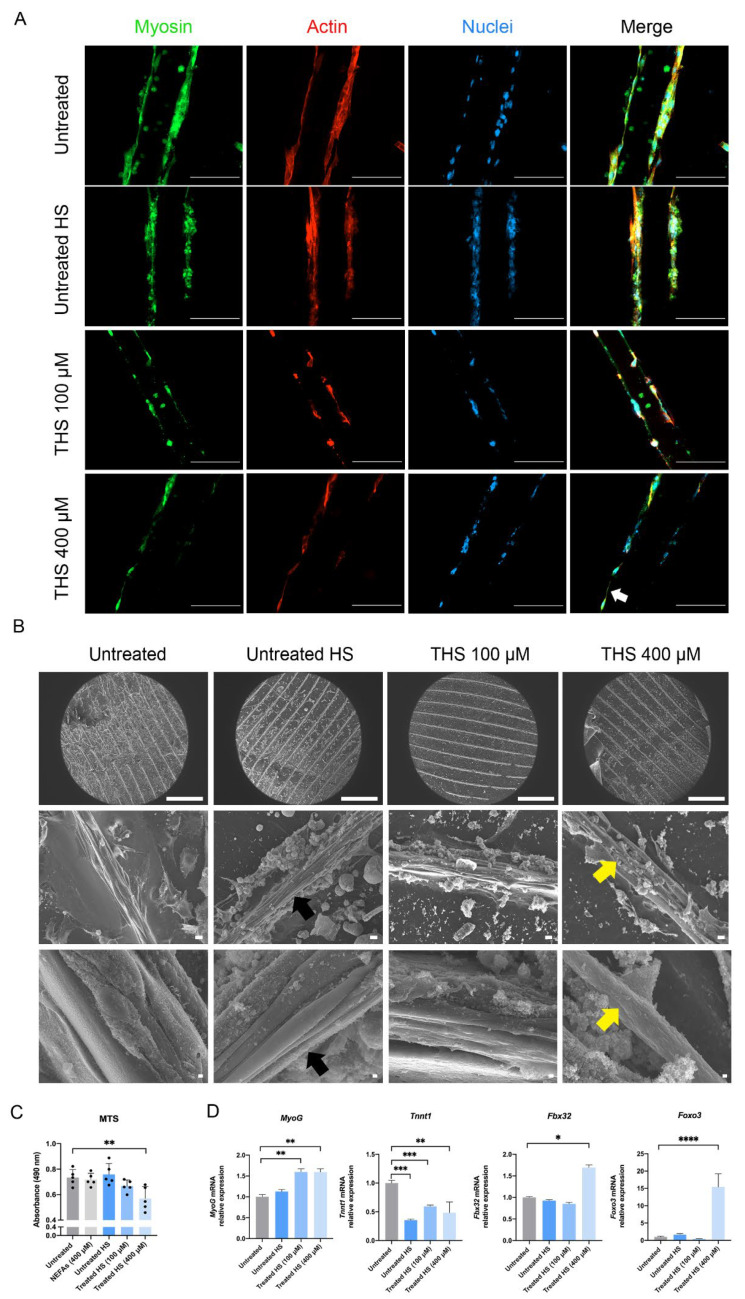
Effect of fatty hepatocytes’ supernatant on 3D skeletal muscle tissues. (**A**) Confocal microscopy of myotubes differentiation markers. White arrow indicates the loss of cytoplasm. Scale bar = 100 µm. Nuclei stained using DAPI. (**B**) Scanning electron microscopy of 3D skeletal muscle tissues incubated with supernatant from fatty hepatocytes. Black and yellow arrows indicate differentiated myotubes and loss of cytoplasm, respectively. Scale bar top panel = 1 mm, middle panel = 10 µm, and bottom panel = 1 µm. (**C**) Metabolic activity of C2C12 myotubes incubated with supernatant from fatty AML12. (**D**) Real-time qPCR of myotubes maturation (MyoG and Tnnt1) and atrophy (Fbx32 and Foxo3) markers. The untreated condition was used as calibrator. The Results are expressed as mean values ± SEM and compared using one-way analysis of variance followed by post hoc tests when appropriate. * *p* < 0.05; ** *p* < 0.01; *** *p* < 0.001; **** *p* < 0.0001. THS: treated hepatocytes’ supernatant; HS: hepatocytes’ supernatant; MTS: 3-[4,5,dimethylthiazol-2-yl]-5-[3-carboxymethoxy-phenyl]-2-[4-sulfophenyl]-2H-tetrazolium; MyoG: Myogenin; Tnnt1: Troponin T1; Fbx32: F-Box Protein 32; Foxo3: Forkhead Box O3.

**Figure 6 biomedicines-10-00958-f006:**
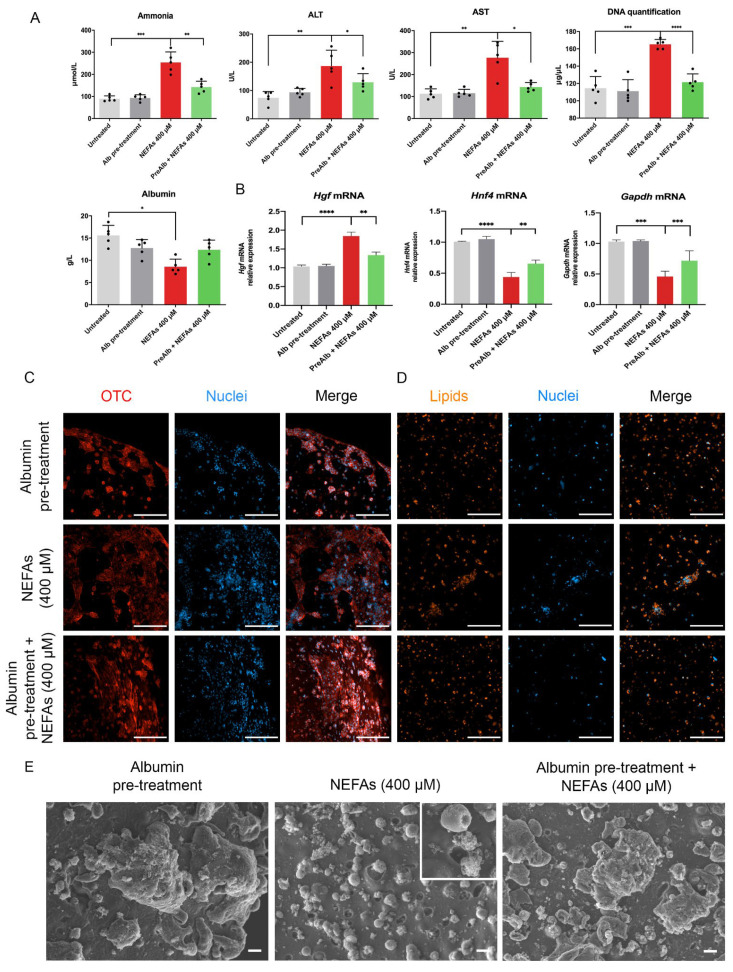
Albumin pre-treatment protect 3D hepatocytes upon challenge with NEFAs. (**A**) Biochemical analyses of the supernatant of 3D AML12 cells pre-treated with albumin and challenged with NEFAs for 72 h assessed by clinical standard procedures. (**B**) Real-time qPCR of hepatocyte markers. The untreated condition was used as calibrator. (**C**) Confocal microscopy of a hepatocyte’s functionality marker of 3D AML12 cells pre-treated with albumin and challenged with NEFAs for 72 h. Scale bar = 200 µm. (**D**) Confocal images of lipids accumulation of 3D AML12 cells pre-treated with albumin and challenged with NEFAs for 72 h in NEFAs treated hepatocytes assessed by AdipoRed™ assay. Scale bar = 200 µm. (**E**) Ultrastructural assessment of AML12 cells phenotype by scanning electron microscopy. Scale bar = 10 µm. The results are expressed as mean values ± SEM and compared using one-way analysis of variance followed by post hoc tests when appropriate. * *p* < 0.05; ** *p* < 0.01; *** *p* < 0.001; **** *p* < 0.0001. NEFAs: non-esterified fatty acids; ALT: Alanine Aminotransferase; AST: Aspartate Aminotransferase; Hgf: hepatocyte growth factor; Hnf4a: hepatocyte nuclear factor 4 alpha; Gapdh: glyceraldehyde 3-phosphate dehydrogenase; OTC: ornithine transcarbamylase.

**Figure 7 biomedicines-10-00958-f007:**
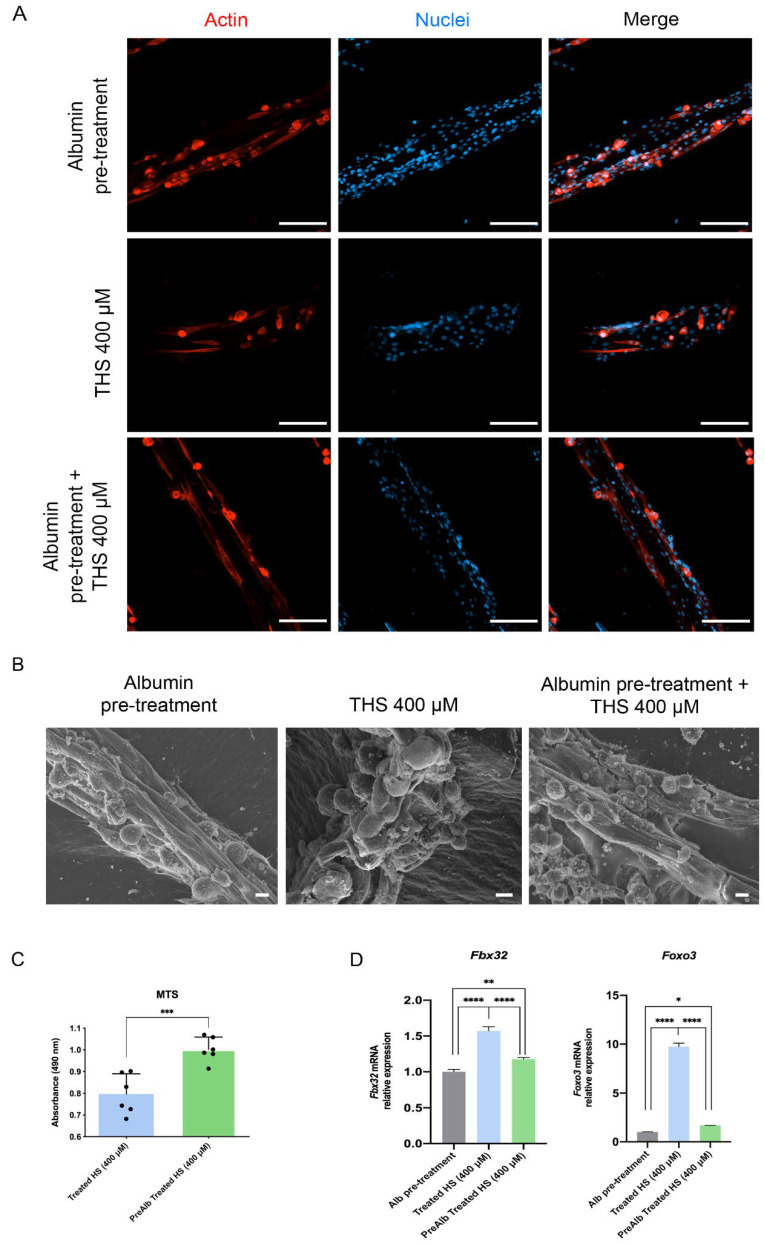
Effect of albumin pre-treated hepatocytes’ supernatant on 3D skeletal muscle tissues. (**A**) Confocal microscopy images of myotubes cytoskeleton. Scale bar = 200 µm. Nuclei stained using DAPI. (**B**) Scanning electron microscopy of 3D skeletal muscle tissues incubated with supernatant from pre-treated hepatocytes. Scale bar = 10 µm. (**C**) Metabolic activity of 3D skeletal muscle tissues incubated with supernatant from pre-treated hepatocytes. (**D**) Real-time qPCR of myotubes atrophy markers. The untreated condition was used as calibrator. The results are expressed as mean values ± SEM and compared using one-way analysis of variance followed by post hoc tests when appropriate. * *p* < 0.05; ** *p* < 0.01; *** *p* < 0.001; **** *p* < 0.0001. THS: treated hepatocytes’ supernatant; MTS: 3-[4,5,dimethylthiazol-2-yl]-5-[3-carboxymethoxy-phenyl]-2-[4-sulfophenyl]-2H-tetrazolium; Fbx32: F-Box Protein 32; Foxo3: Forkhead Box O3.

## Data Availability

Not applicable.

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
