# Peer review of "Fatty Hepatocytes Induce Skeletal Muscle Atrophy In Vitro: A New 3D Platform to Study the Protective Effect of Albumin in Non-Alcoholic Fatty Liver"

_biomedicines, 2022, doi:10.3390/biomedicines10050958_

Round 1

Reviewer 1 Report

This study describes an in-vitro model to examine the role of fatty acid overloaded hepatocytes on sarcopenia.

Title; please use lower case letter for “Non-alcoholic”

There are numerous grammatical and typing errors and the manuscript has to be carefully corrected by a native speaker.

Is sarcopenia common in patients with fatty liver or just those with NASH?

Does the 3D model represent fatty liver or NASH?

type 2 diabetes mellitus (T2DM) - Diabetes mellitus Type 2 (DMT2) – please use one description.

„sarcopenia is a strong predictor of poor post-operative outcomes” all surgerys?

“ foetal bovine serum“ should be “fetal”

10 mg/ml insulin, is this correct?

Figure 1; please explain abbreviations in the legend and not in the figure.

“C2C12 (,“ “Whitin“ please correct

„supplemented with 10% foetal bovine serum (FBS; ATCC 30-2020)” is explained and defined two times, lane 85 and 90

e.g. 107, please use upper case

“Previous experiments were performed to establish that optimal  cells number for each tissue” has this been published?

“The cells were mixed with the GelMA-CMCMA mixture were mixed” please correct.

“Palmitate acid (PA) (P0500, Sigma) solution was prepared adding 1 g of PA to 9,75 mL of  100% isopropanol” above authors refer to palmitic acid. 9,75 is 9.75.

“for 3 hours at 37oC,“ “to isolate total mRNA cells “ please correct.

Unclear description, please explain: “a 6-4 ratio“ “4 degrees“ ?

 “rhodamine-phalloidin“ is not an antibody.

Total RNA and not mRNA was purified. Please correct.

“reduces 273 the metabolic activity“ which ones?

“at all time points (p<0.01, p<0.001, and 0.0541, respectively” refer to which time point?

Figure 2G, what do the arrows mean?

“As expected, the  MTS assay showed higher metabolic activity in the 150 μL hydrogel compared with the  50 μL hydrogel” this needs some explanation.

“For further experiments, we have chosen 50 μL as the  volume of encapsulation because of the higher live/dead cells ratio compared with and  not 150 μL” correct, please carefully correct the whole manuscript.

“This section of the study demonstrates the beneficial impact of  the 3D structure on the cellular phenotype.“ This is unclear. Do 2D cells or normal cultured cells express these markers?

“red, yellow, and black  arrows, respectively, Figure 4B” explanation should be moved to the figure legends and deleted in the text.

Figure 4E, does the untreated 3D model cause higher levels of DNA?

“Finally, the 3D models  generated higher levels of markers than 2D models meaning” this is not correct for ALT and albumin.

“The MyoG gene increases while Tnnt1 gene expression decreased significantly upon NE-394 FAs treatment” Tnnt1 is already low in untreated HS treated cells. This may not be related to NEFAs.

A further limitation is that skeletal muscle cell line was used.

Can the authors exclude that PA/OA alone affect the function of C2C12 cells?

“We found a high level of Hgf in a  similar way that it has been found increased in NASH patients. In the presence of oxidative stress due to a high fat regimen, Hgf has an antioxidant response GapdhHgfHgf[23].” First sentence may need a reference, Words before reference 23 should be deleted.

Supplementary figure 1, abbreviations should be explained in the legend.

Supplementary figure 3: please explain arrows.

Supplementary figure 4a: Is this mRNA or protein expression?

Author Response

Review 1

We want to thank you the reviewer for taking the time to read our manuscript and provide constructive feedback. Below, we have included a point-by-point answer below. Our responses are highlighted in blue.

Title; please use lower case letter for “Non-alcoholic”

Done.

There are numerous grammatical and typing errors and the manuscript has to be carefully corrected by a native speaker.

We provide a revised version of the manuscript.

Is sarcopenia common in patients with fatty liver or just those with NASH?

Sarcopenia is present in 18%-38% of NAFLD patients, and 35%-63% with NASH. Those with sarcopenia also had higher rates in NASH with fibrosis (46%) versus those without (25%), with a 2.5-fold higher risk for NASH and significant fibrosis in patients with NAFLD, independent of obesity and insulin resistance (Koo BK, et al. Sarcopenia is an independent risk factor for non-alcoholic steatohepatitis and significant fibrosis. J Hepatol 2017;66:123-131. Lee YH, et al. Sarcopaenia is associated with NAFLD independently of obesity and insulin resistance: nationwide surveys (KNHANES 2008-2011). J Hepatol 2015;63:486-493)

Does the 3D model represent fatty liver or NASH?

Our model mimics the NAFLD phenotype. NASH is characterized by an extensive inflammatory response driven by other liver cell type such as Hepatic stellate cells and Kupffer cells, not included in this study.

type 2 diabetes mellitus (T2DM) - Diabetes mellitus Type 2 (DMT2) – please use one description.

Done.

„sarcopenia is a strong predictor of poor post-operative outcomes” all surgerys?

We have specified “sarcopenia in turn is a strong predictor of poor post-liver transplant outcomes” within the text.

“ foetal bovine serum“ should be “fetal”

Done.

10 mg/ml insulin, is this correct?

No. Thank you to the reviewer for spotting this out. We corrected the error: “10 μg/ml”

Figure 1; please explain abbreviations in the legend and not in the figure.

Done.

“C2C12 (,“ “Whitin“ please correct

Done.

„supplemented with 10% foetal bovine serum (FBS; ATCC 30-2020)” is explained and defined two times, lane 85 and 90

We removed one.

e.g. 107, please use upper case

Done.

“Previous experiments were performed to establish that optimal  cells number for each tissue” has this been published?

No, it did not. We added the formula “data not shown”.

“The cells were mixed with the GelMA-CMCMA mixture were mixed” please correct.

Done.

“Palmitate acid (PA) (P0500, Sigma) solution was prepared adding 1 g of PA to 9,75 mL of  100% isopropanol” above authors refer to palmitic acid. 9,75 is 9.75.

Done.

“for 3 hours at 37oC,“ “to isolate total mRNA cells “ please correct.

Done.

Unclear description, please explain: “a 6-4 ratio“ “4 degrees“ ?

Thank you to the reviewer for pointing this out. We have now rectified the information at line 407.

 “rhodamine-phalloidin“ is not an antibody.

We have now rectified the information at line 483.

Total RNA and not mRNA was purified. Please correct.

Done.

“reduces 273 the metabolic activity“ which ones?

Done.

“at all time points (p<0.01, p<0.001, and 0.0541, respectively” refer to which time point?

Done.

Figure 2G, what do the arrows mean?

We have now added the information in the figure legend.

“As expected, the  MTS assay showed higher metabolic activity in the 150 μL hydrogel compared with the  50 μL hydrogel” this needs some explanation.

Thanks to the reviewer for the observation. We agreed that the sentence was misleading. We referred to higher metabolic in the 150 μL hydrogel compared with the 50 μL hydrogel because of the more cells present. The cells number is the same in both conditions.

“For further experiments, we have chosen 50 μL as the  volume of encapsulation because of the higher live/dead cells ratio compared with and  not 150 μL” correct, please carefully correct the whole manuscript.

Thanks again to the reviewer. We have now corrected the whole manuscript.

“This section of the study demonstrates the beneficial impact of  the 3D structure on the cellular phenotype.“ This is unclear. Do 2D cells or normal cultured cells express these markers?

We agree with reviewer. The sentence was misleading. We rephrased the sentence: line 273 “The hydrogel allows the long-term culture of the cells without affecting their phenotype, unfeasible using 2D culture.”.

“red, yellow, and black  arrows, respectively, Figure 4B” explanation should be moved to the figure legends and deleted in the text.

Done.

Figure 4E, does the untreated 3D model cause higher levels of DNA?

Compared with 2D, yes, it is. The reason might be the higher number of the encapsulated cells within the hydrogel compared with 2D as visible in figure 3D using live/death assay.

“Finally, the 3D models  generated higher levels of markers than 2D models meaning” this is not correct for ALT and albumin.

We have now rectified the information. Please, see line 719.

“The MyoG gene increases while Tnnt1 gene expression decreased significantly upon NE-394 FAs treatment” Tnnt1 is already low in untreated HS treated cells. This may not be related to NEFAs.

We completely agree with the reviewer. We cannot explain this results at moment.

A further limitation is that skeletal muscle cell line was used.

We agree with the reviewer. 

Can the authors exclude that PA/OA alone affect the function of C2C12 cells?

We did not observe any effect on the metabolic activity in the C2C12 cells upon PA/OA alone treatment (Figure 5C).

“We found a high level of Hgf in a similar way that it has been found increased in NASH patients. In the presence of oxidative stress due to a high fat regimen, Hgf has an antioxidant response GapdhHgfHgf[23].” First sentence may need a reference, Words before reference 23 should be deleted.

Thanks to the reviewer for pointing this out. We have now corrected the issues. Please see line 918. 

Supplementary figure 1, abbreviations should be explained in the legend.

Done.

Supplementary figure 3: please explain arrows.

Done.

Supplementary figure 4a: Is this mRNA or protein expression?

We have now rectified the information.

Reviewer 2 Report

In this work, De Chiara and coworkers investigated the 2D/3D in vitro cross-talk between skeletal muscle cells and NEFAs-exposed hepatocytes mimicking NAFLD disease.

Although with some limitations, correctly stated in the discussion section (i.e. the use of mouse cells, the absence of stellate and satellite cells, the lack of in vivo experiments), the work is well organized and contains several data worth to be published. Just three minor suggestions are reported below.

  • Please state also in the introduction section the reasons of choice of GelMA and CMC hydrogels, as in the discussion section (lines 437-440)
  • Correct typos throughout the manuscript
  • Provide a graphical abstract to depict the main results of the work.

Author Response

Review 2

In this work, De Chiara and coworkers investigated the 2D/3D in vitro cross-talk between skeletal muscle cells and NEFAs-exposed hepatocytes mimicking NAFLD disease.

Although with some limitations, correctly stated in the discussion section (i.e. the use of mouse cells, the absence of stellate and satellite cells, the lack of in vivo experiments), the work is well organized and contains several data worth to be published. Just three minor suggestions are reported below.

We are grateful to the reviewer for the constructive comments.

  • Please state also in the introduction section the reasons of choice of GelMA and CMC hydrogels, as in the discussion section (lines 437-440)

Thanks to the reviewer for pointing this out. We have now added the missing information. Please see line

  • Correct typos throughout the manuscript

We have our manuscript revised for typos error.

  • Provide a graphical abstract to depict the main results of the work.

We have now added the graphical abstract in Figure 1A. 

Round 2

Reviewer 1 Report

The authors refer to NAFLD and NASH. However, NAFLD covers fatty liver and NASH. It seems that the model is representative of non-alcoholic fatty liver. This has to be clearly described in the paper. Prevalence of sarcopenia in non-alcoholic fatty liver and NASH should be described in the introduction of the paper. 

Author Response

Once again we thank to the reviewer for the observation. We agree that it was misleading for the reader.The changes are in red.

We changed NAFLD for NAFL throughout the whole manuscript.

We added the prevalence of Sarcopenia in the introduction as suggested by the reviewer.

Regards,